# A Fluidics-Based Double Flexural Membrane Piezoelectric Micromachined Ultrasonic Transducer (PMUT) for Wide-Bandwidth Underwater Acoustic Applications

**DOI:** 10.3390/s21165582

**Published:** 2021-08-19

**Authors:** Khairul Azman Ahmad, Mohamad Faizal Abd Rahman, Khairu Anuar Mohamed Zain, Muhammad Naim Haron, Asrulnizam Abd Manaf

**Affiliations:** 1Collaborative Microelectronic Design Excellence Center (CEDEC), Universiti Sains Malaysia, Sains@USM, Bayan Lepas 11900, Pulau Pinang, Malaysia; khairuluitm75@gmail.com (K.A.A.); faizal635@uitm.edu.my (M.F.A.R.); anuar@usm.my (K.A.M.Z.); naimharon@student.usm.my (M.N.H.); 2School of Electrical Engineering, College of Engineering, Universiti Teknologi MARA, Cawangan Pulau Pinang, Kampus Permatang Pauh, Bukit Mertajam 13500, Pulau Pinang, Malaysia

**Keywords:** double flexural structure, fluidic backing layer, wide-band high-sensitivity underwater acoustic transducer, printed circuit board

## Abstract

In acoustic receiver design, the receiving sensitivity and bandwidth are two primary parameters that determine the performance of a device. The trade-off between sensitivity and bandwidth makes the design very challenging, meaning it needs to be fine-tuned to suit specific applications. The ability to design a PMUT with high receiving sensitivity and a wide bandwidth is crucial to allow a wide spectrum of transmitted frequencies to be efficiently received. This paper presents a novel structure involving a double flexural membrane with a fluidic backing layer based on an in-plane polarization mode to optimize both the receiving sensitivity and frequency bandwidth for medium-range underwater acoustic applications. In this structure, the membrane material and electrode configuration are optimized to produce good receiving sensitivity. Simultaneously, a fluidic backing layer is introduced into the double flexural membrane to increase the bandwidth. Several piezoelectric membrane materials and various electrode dimensions were simulated using finite element analysis (FEA) techniques to study the receiving performance of the proposed structure. The final structure was then fabricated based on the findings from the simulation work. The pulse–echo experimental method was used to characterize and verify the performance of the proposed device. The proposed structure was found to have an improved bandwidth of 56.6% with a receiving sensitivity of −1.8864 dB rel 1 V µPa. For the proposed device, the resonance frequency and center frequency were 600 and 662.5 kHz, respectively, indicating its suitability for the targeted frequency range.

## 1. Introduction

For underwater applications, PMUTs are widely used in different areas, including for control, communication, and imaging. Different applications require different bandwidths, which can generally be classified into three main frequency regions: low frequencies below 100 kHz, medium frequencies ranging from 100 kHz to 1 MHz, and high frequencies above 1 MHz.

The use of PMUTs as underwater communication receivers began to attract attention due to their high sensitivity for medium-range frequencies [1]. PMUT studies are often based on the polarization of piezoelectric films, which generate the electric charge required to induce an electrical signal in response to the amount of acoustic energy hitting the membrane [2]. Two polarization modes are normally adapted for this piezoelectric effect, i.e., thickness and in-plane modes. The implementation of these two modes with their different performance parameter is achieved by implementing different electrode and membrane structures [3,4,5,6]. The sensitivity of the polarization method, which involves an acoustic transducer with an in-plane mode, allows better performance compared to the thickness-mode polarization method [7,8,9].

For a PMUT that is to be used as a receiver, two parameters are important in its design, namely bandwidth and receiving sensitivity. These parameters are very much dependent on the structural design and materials. Previous studies have shown how different structures were implemented to give different performances in order to suit targeted applications [10,11]. The piezoelectric membrane is the main sensing element that requires attention in every PMUT design. Different membrane shapes have been proposed to investigate the sensitivity and bandwidth characteristics of ultrasonic transducers [4,8,12]. Circular- and square-shaped membranes were among the popular shapes chosen for PMUT. By varying the shapes and dimensions of the membrane, such as the thickness and size, different performance outputs can be obtained. Additionally, the choice of material selected as the membrane also affects the performance of a PMUT due to the acoustic strain relationship, which is determined by the piezoelectric constant [13,14]. Common piezoelectric materials such as zinc oxide (ZnO), lead zirconate titanate (PZT), and aluminum nitrate (AlN) are among the prevalent membrane materials, having different coupling coefficients and dielectric constants, which can be manipulated to produce different performance results [4,8,13,14]. The selection of different membrane shapes, dimension sizes, and suitable materials depends on the complexity of the process and the fabrication technique, as well as the available facilities.

The use of a cavity also has a significant effect on the performance of a PMUT. Cavity-based devices are preferred, as they provide space for membrane deflection, increasing the sensitivity [3]. The use of additional layers such as matching and backing layers has also been proposed in several studies for specific applications [6,14]. Matching layers are crucial in reducing the acoustic impedance between different transmission mediums, while the backing layers inside cavities have been proposed by some researchers to withstand excessive deflection and avoid membrane rupture. The backing layer may also be used to dampen the ringing effect [15].

All structural variations impact the performance of a PMUT, either in terms of the receiving sensitivity or frequency bandwidth [16]. The trade-off between the bandwidth and receiving sensitivity is crucial during the design stage in order for a device to suit a specific application [12,17,18]. In this study, a novel structure involving a double flexural membrane with a backing layer is proposed to develop a PMUT that can act as an acoustic receiver in the wide-band, medium-frequency operation range of 300 to 800 kHz. This study involves the structural design and fabrication of the device, followed by testing to verify the simulations.

## 2. Background Theory and Proposed Design

Figure 1a shows the overall structure of the proposed device. This consists of a double membrane and double cavities, aiming to provide both good sensitivity and a wide bandwidth for the targeted application. The proposed structure adopts an in-plane polarization structure to induce more charge during acoustic detection.

The device is basically divided into two main functional layers, namely the first and second flexural layers. Under each flexural layer there is a cavity, namely an air cavity or fluidic cavity. The first layer of the flexural structure consists of an acoustic matching layer (PDMS), first flexural membrane (PZT5H), air cavity, and electrodes. In theory, when an ultrasonic wave hits the PMUT, the first flexural membrane bends and induces a charge at the positive electrode. The penetration of the ultrasonic wave from the first flexural membrane propagates within the air cavity and bends the second flexural membrane (polyimide) as shown in Figure 1b. The thickness of the air cavity is not influenced by the increase in stress that occurs in the PZT5H membrane. The presence of the air cavity improves the stress inside PZT5H membrane. Some of the ultrasonic waves penetrate the polyimide and propagate into the second flexural layer, which contains castor oil and PDMS. The second flexural layer consists of castor oil, which works as a fluidic backing layer in its cavity; the PDMS layer functions as a fluidic backing container and as its substrate. The wave bends the fluidic backing container (PDMS) and creates a double stress capacity to the membrane, thus inducing more charge. In the proposed design, the backing layer accommodates a suitable fluidic material for the purpose of strengthening the membrane against hydrostatic pressure (for underwater application). It also gives the effect of a mechanical damper that absorbs the ultrasonic signal pulse energy; hence, it shortens the signal, thereby widening the transducer frequency bandwidth.

## 3. Methodology

### 3.1. Modeling and Simulation

COMSOL Multiphysics 5.0 was used to perform the finite element method (FEM) simulations in order to evaluate the receiving performance of the proposed PMUT design. In general, the PMUT design is targeted as an acoustic receiver; the performance is simulated by an open-circuit receiving response (OCRR) and receiving sensitivity (RS). In COMSOL, the open-circuit receiving response (OCRR) is obtained from an integral part of the normal current density over the electrostatic angular frequency and the capacitance of the piezoelectric material, as given by Equation (1) [19]:(1)$$\mathrm{OCRR}=\frac{int\;op\left(es.JZ\right)}{\left(es.omega*C\right)}$$
where “*int op*” is the integral function, *es* is the electrostatic setup, *JZ* is the current displacement, *es.omega* is the impedance, and *C* is the capacitance of the piezoelectric membrane.

The settings for all parameters are included in the geometric entity setup program, while the OCRR was measured in V/μPa. The RS can be obtained from the logarithmic function of the OCRR and is measured in dB relative to 1 V/μPa, as given by Equation (2) [20]:(2)RS=20log10(OCRR)

Both the OCRR and RS were simulated to obtain the relationship between pressure and voltage, which corresponds to the performance of the proposed PMUT as an acoustic receiver. By varying the PMUT structural design parameter, the simulation is divided into two parts based on different membrane materials and different electrode distance settings. Both tasks were performed as a guide for fabrication of the final device.

### 3.2. Different Materials of the Membrane (First Flexural Structure)

This task aimed to study the effects of the different membrane materials that influence the receiving sensitivity of the acoustic signal. For simplicity, only the first flexural structure was considered in this task. Figure 2 shows the functional structures considered in the simulation to evaluate the first flexural structure.

The air cavity shown in Figure 2 is used to increase the receiving sensitivity, as explained in the previous section. The two electrodes on the right and left of the membrane are the positive electrode and negative electrode, respectively. The flexural mode is formed when mechanical oscillation acts on the membrane. PDMS is introduced on top as a matching layer to reduce the acoustic impedance between the transmission medium and the membrane, improving the receiving capabilities of the device. Table 1 summarizes the dimensions of the structure used for the simulation, describing all of the lengths, widths, and thicknesses for the electrodes, PDMS (matching layer), membrane, and polyimide (substrate).

Initially, the simulation was performed to analyze the receiving performances of different membrane materials. Five different materials were selected for the simulation. Pressure equaling 1 MPa was applied to the top of the membrane and the frequencies varied between 25 and 1500 kHz, with steps of 25 kHz. Table 2 shows all of the parameters and properties (Young’s modulus, mass density, dielectric constant, Poisson ratio, acoustic impedance, and piezoelectric coefficient (*d*_33_)) of the piezoelectric materials simulated in this study. Mass density, *ρ* for aluminum nitrate was around 3250 to 3300. In simulation, mass density was depicted of 3300 for maximum value due to obtain minimum sensitivity of transducers.

Based on these specifications, the OCRR was simulated using COMSOL for the specified range of the frequencies.

### 3.3. Electrode Distances

The second FEA simulation was performed to investigate the influence of the electrode dimensions on the receiving capabilities and to demonstrate the effects of the distance between the electrodes on the open-circuit receiving response (OCRR). In this task, PZT was selected as the membrane due to its high electromechanical coupling coefficient and resonance frequency occurring in the targeted frequency range. By using the same model as in Figure 2 and the same dimensions as in Table 1, the electrode distance was varied from 0.1 to 1.0 mm with steps of 0.2 mm. The tasks were significant, as the distance between electrodes influences the output voltage. Based on theory, the output voltage can be increased if the electrodes are increased; however, there are other considerations to be taken into account, such as the fabrication complexity and the suitability of the gap to hold the membrane. All frequencies and pressure setups are maintained as previously described.

### 3.4. Double Flexural Membranes and Cavities with Fluidic Backing Layers

Based on the findings from Section 3.2 and Section 3.3, the best membrane and electrode distances were then finalized as the targeted dimensions for the final device. Additional layers, namely PDMS and backing layers (castor oil), were included in this final model to act as an acoustic matching layer and a liquid backing layer, respectively. The thickness of the matching layer was based on the equation shown in Equation (3) [20]:(3)$$Z_{M}=\sqrt{Z_{W}Z_{PZT}}$$
(4)$$Z_{M}=\rho \nu$$
(5)ν=λf
(6)λ=2t
where *Z_M_* is the acoustic impedance of the matching layer, *Z_W_* is the acoustic impedance of water, and *Z_PZT_* is the acoustic impedance of lead zirconate titanate. Equation (4) calculates the acoustic impedance, which can be calculated from the density, *ρ*, and sound velocity, *v*, of the matching layer. Equation (5) is a wavelength equation used for impedance matching, where *v* is the sound velocity of the matching layer and *f* is the frequency of the matching layer [21]. The wavelength equation can also be calculated as twice the matching layer thickness [22]. Finally, the simulation was carried out based on the same previously explained procedure to obtain and analyze the OCRR of the final device. Figure 3 shows the schematic diagram of the double-membrane structure with the simulated backing layer.

A fluidic layer is a new type of backing layer proposed in this work that can be used for targeted underwater applications. The layer can be applied to the proposed PMUT device as a second cavity structure to improve the bandwidth of the PMUT device. A fluidic backing layer is a mechanical damper that absorbs the ultrasonic signal pulse energy and shortens the signal, widening the transducer frequency bandwidth. Castor oil is used as a fluidic material as it can protect the PMUT membrane from hydrostatic pressure. Table 3 summarizes the targeted dimensions of the final proposed model by including the castor oil in the second structural cavity as the backing layer.

The simulated OCRR will be compared to the results for the single-layer membrane structure to study the effects of introducing the backing layer (castor oil). The same structure needs to be fabricated and tested experimentally (to obtain OCRR) to verify the simulation work.

### 3.5. Device Fabrication

Based on the model simulated previously, the final device will be fabricated to verify the performance of the proposed structure. Figure 4 illustrates the process flow used to fabricate the final structure of the proposed device. Castor oil was injected into the backing layer container with a full load of 4.45 mm^3^. The backing layer was constructed with a λ/4-thick matching layer [23,24]. The thickness of the backing layer must more than four times that of the wavelength matching layer.

### 3.6. Experimental Setup for OCRR Measurement of the Fabricated PMUT

This study was performed to evaluate the performance of the proposed device based on the OCRR measurements. The results will be compared with the previously simulated OCRR. Figure 5 shows the initial experimental setup, which consisted of a test tank, a projector (UNDT-500 kHz, Technotronics Industries, Delhi, India), a pulse-forming network (PFN), a function generator, a picoscope, and a laptop.

At first, a burst signal is created using a pulse-forming network and function generator. The function generator (Hameg HM-8150) is set at 10 V peak-to-peak sinusoidal voltage with a frequency setting starting at 25 kHz and ranging to 1500 kHz, with increasing steps of 25 kHz. The pulse-forming network is programmed to provide a pulse square wave signal, which is modulated with a sinusoidal voltage from the function generator. A commercial projector and hydrophone are initially calibrated to ensure the reliability of the measured and collected data. After calibration, the hydrophone is replaced by the fabricated PMUT. Ping signals are used in this experiment to identify and investigate the performance of the designed PMUT underwater. The pulse–echo method is used to measure the open-circuit receiving response. The picoscope is used to collect the input and output data from the projector and the PMUT, and these data are displayed and stored on the laptop. Based on the data, the OCRR obtained through this experiment is compared with that obtained through the simulation in order to verify the simulation results.

## 4. Result and Discussion

This section presents the results obtained from simulation and experimental studies to observe the open-circuit receiving response (OCRR) performances of different membrane materials and electrode distances.

### 4.1. Membrane

Figure 6 shows the OCRR simulation results for a single flexural membrane at a frequency range of 0 to 1500 kHz. Five different membrane materials were used in this simulation work. In general, it can be observed that the OCRR values fluctuate throughout the frequency range of interest. Membrane materials also indicate the possibility of having multiple resonance frequencies throughout the range of simulated frequencies due to the rectangular shape of the electrodes and the flexural-mode membrane, which contributes to the multiple resonance frequency modes. Out of these, the best two materials that show good receiving responses are AIN and PZT5H. The material properties of AIN and PZT5H, such as the piezoelectric constant and dielectric properties, contribute to their high OCRR values. AIN exhibits two resonance frequency points at 1100 and 1300 kHz with OCRR values of −0.083 and −0.003 dB, respectively.

PZT5H shows two resonance frequency points at 475 and 575 kHz, with OCRR values of −0.911 and −0.882 dB, respectively. For the targeted application, PZT5H was found to be more suitable for use in the final design because its resonance frequencies lay within the targeted range of 400 to 800 kHz. Based on Figure 6, a narrow bandwidth can also observed for all membrane materials used for the simulated structure.

### 4.2. Electrodes Distance

Figure 7 shows the OCRR for a single flexural PZT5H membrane. The use of PZT5H was due to its high electromechanical coupling coefficient and resonance frequency occurring in the range of 400 to 800 kHz, as shown in Figure 6.

The simulation results demonstrate the effects of the distances between electrodes on the OCCR. The OCCR represents the receiving sensitivity of the PMUT. In this study, we aimed to find the best electrode distance that could give a resonance frequency within the targeted frequency range. It was observed that the highest receiving sensitivity was −0.33526 dB rel 1 V/µPa, with a resonance frequency of 1225 kHz, at an electrode distance of 1.0 mm. The second-highest receiving sensitivity was from the structure with an electrode distance of 0.8 mm and a receiving sensitivity of −0.53707 dB rel 1 V/µPa, which occurred at a frequency of 1325 kHz. Next, the PMUT with distances between the two electrodes of 0.5 mm gave a resonance frequency of 575 kHz, with a receiving response of −0.882 dB rel 1 V/µPa. Since the resonance frequency at 0.5 mm lay within the targeted frequency range, this dimension gap was selected as a parameter for the final device.

### 4.3. Simulation of Single-Membrane (No Backing Layer) and Double-Membrane Structures (Added Fluidic Backing Layer)

Figure 8 shows the simulated OCRR for single- and double-membrane PMUT structures. Both contained the same membrane material, PZT5H, as well as the same electrode distance, as determined previously. The aim of this analysis was to observe the improvements in bandwidth for both structure types.

For the double structure, castor oil was used as the backing layer, which gives a damping effect and can improve the bandwidth of the device’s receiving performance. Observations were made at a resonance frequency of around 575 kHz for both PMUTs; the OCRR of the PMUT without a backing layer was 0.88 dB rel 1 V/µPa, higher than that of the PMUT with a backing layer, which was −4.56 dB rel 1 V/µPa. This indicates that by adding a backing layer, the receiving sensitivity was somewhat reduced; however, in terms of the bandwidth, the dual flexural layers provided an improved bandwidth, which was suitable for wide-band applications. As can be observed in Figure 9, the single-membrane structure shows a fractional bandwidth of 9.1%, while the double-membrane structure shows a fractional bandwidth of 56.6%. The fractional bandwidth can be calculated using Equation (7) [25]:(7)$$\mathrm{Fractional}\;\mathrm{bandwidth}\;=\frac{\left(f_{u}-f_{l}\right)}{\left(f_{u}+f_{l}\right)}\times \;2\;\times \;100\%$$
where *f_u_* is the upper frequency and *f_l_* is the lower frequency at −6 dB.

This shows that the proposed double-membrane structure (with a castor oil backing layer) is capable of extending the bandwidth from 9.1% (single flexural membrane structure fractional bandwidth) to 56.6% (dual membranes with backing layer). This finding proves the significance of adding the fluidic backing layer (castor oil), which creates a wider bandwidth; hence, this finding is in agreement with our initial hypothesis.

### 4.4. OCRR for the Final PMUT Device (Simulated vs. Experimental Results)

Figure 9 shows the open-circuit receiving response (OCRR) results for the simulated and experimental double-membrane PMUTs.

In Figure 9, it can be observed that the response pattern for both methods are similar, proving the validity of the methodology adapted in this work, especially the simulation process; however, some of the experimental results show higher receiving response points than for the simulation results. This is possibly due to the ideal condition assumptions made in the simulation environment, involving perfectly matching conditions and perfectly converted mechanical-to-electrical energy. The calculations are also based on a confined space with no disturbance assumed to be affecting the transmitted and received signals. Additionally, the mechanical properties, such as the Young’s modulus, Poisson’s ratio, and density, are also possible factors that affected the calculation of the receiving responses. In terms of the receiving performance, it can be observed that the proposed PMUT structure has a bandwidth of 56.6% with a center frequency of 662.5 kHz. The resonance frequency for the experimental results was 600 kHz, while the receiving sensitivity was −1.8864 dB rel 1 V/µPa; for the simulation results, the resonance frequency was 575 kHz and the receiving sensitivity was −4.3394 dB rel 1 V/µPa. The results showed that the bandwidth was improved from 9.1% to 56.6% when a fluidic backing layer was implemented in the device. The receiving sensitivity was also improved to −1.8864 dB rel 1 V/µPa when the double-layer method was applied.

## 5. Conclusions

Throughout this study, a novel structure involving dual flexural membranes with a backing layer was successfully fabricated and tested. The device is meant to be used as an acoustic receiver, providing the advantage of a wider bandwidth at an acceptable receiving sensitivity. The wide bandwidth is suitable for underwater applications, and can work across a wide operating frequency range of 475 to 800 kHz. At this frequency range, the receiving sensitivity ranges from −8.5 dB to −1.8 dB. The device’s high receiving sensitivity can eliminate noise due to the high power intensity. The double-flexural method used to design the device was successfully shown to offer improvements in receiving sensitivity, while the fluidic bandwidth and matching layer improved the bandwidth. The final test verified the simulations performed in the earlier design stages and proved the reliability of the OCRR predictions used for the acoustic design. This study also shows several possible design areas that can be further explored to produce acoustic devices with different specifications and performance parameters. The structural configuration, dimensions, and material properties are areas that can possibly affect the performance of a device in terms of its bandwidth and sensitivity. The device achieved a receiving sensitivity of −1.8864 dB rel 1 V/µPa, resonance frequency of 600 kHz, and fractional bandwidth of 56.6%. In conclusion, based on the obtained results, the objectives of this study were successfully achieved.

## Figures and Tables

**Figure 1 sensors-21-05582-f001:**
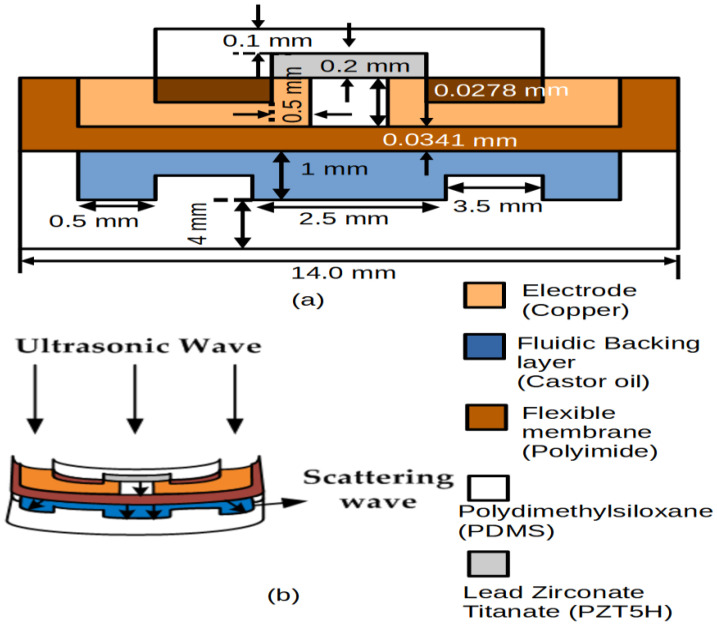
Diagram of the proposed structure of the double flexural membrane sensing device (**a**) and penetration of the ultrasonic wave from the first flexural membrane to second flexural membrane (**b**).

**Figure 2 sensors-21-05582-f002:**
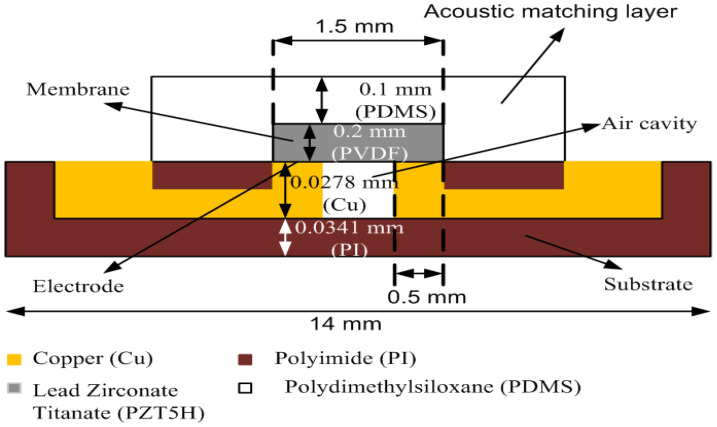
Schematic drawing of the new structure proposed for in-plane polarization.

**Figure 3 sensors-21-05582-f003:**
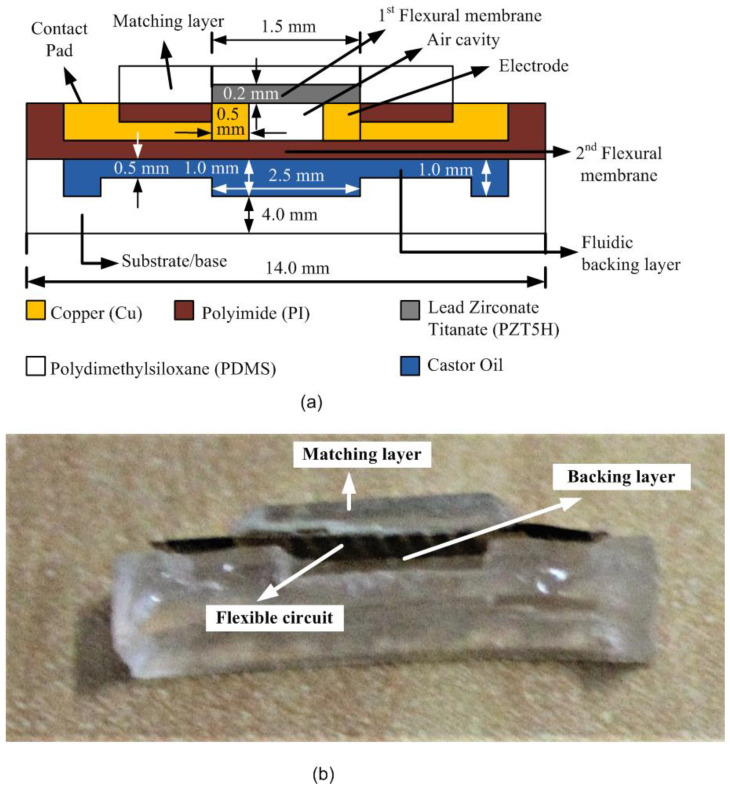
A cross-sectional view of a fluidics-based, double-membrane piezoelectric micromachined ultrasonic transducer: (**a**) schematic diagram; (**b**) digital microscope computer image.

**Figure 4 sensors-21-05582-f004:**
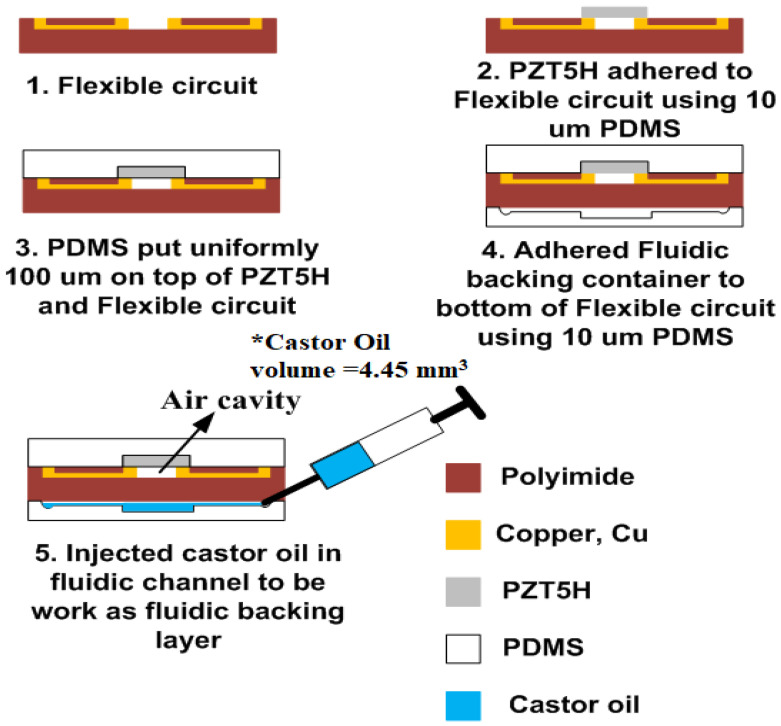
Fabrication process flow of PMUT.

**Figure 5 sensors-21-05582-f005:**
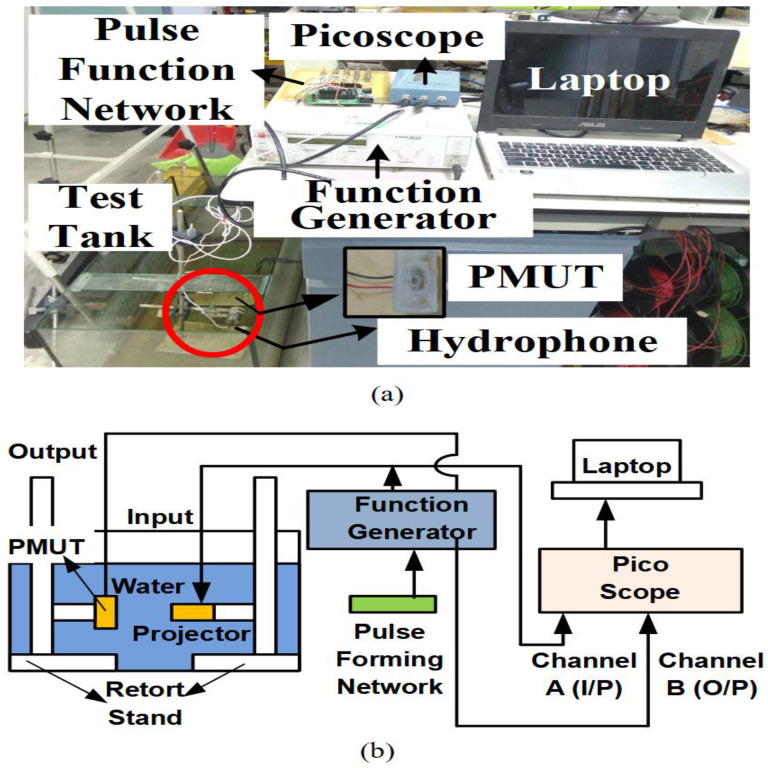
Pulse–echo experimental setup: (**a**) experimental field-work setup; (**b**) schematic drawing of the pulse–echo experimental setup.

**Figure 6 sensors-21-05582-f006:**
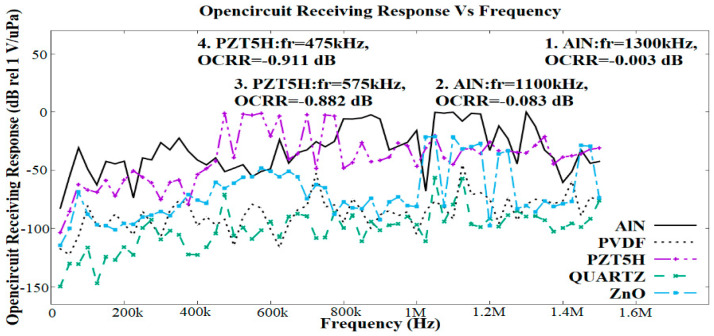
FEA simulation results regarding the effects of the piezoelectric material membranes, such as AlN, PVDF, PZT5H, quartz, and ZnO, on the PMUT in terms of the open-circuit receiving response (OCRR) vs. frequency.

**Figure 7 sensors-21-05582-f007:**
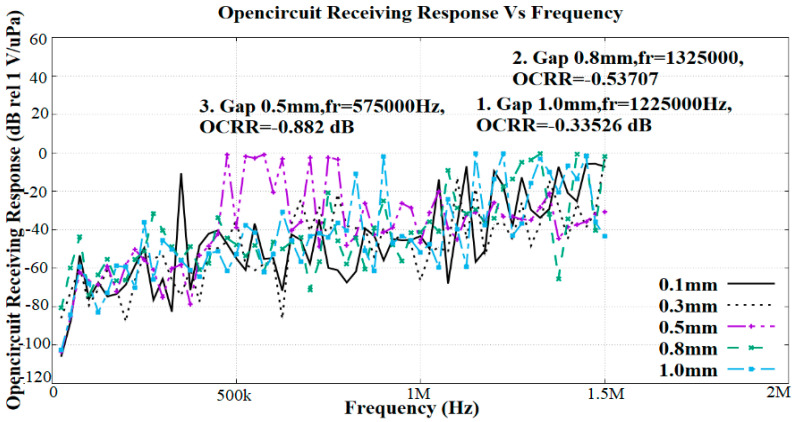
FEA simulation results for the effects of the distance between two electrodes towards in-plane polarization of the PMUT in terms of the open-circuit receiving response (OCRR) vs. frequency.

**Figure 8 sensors-21-05582-f008:**
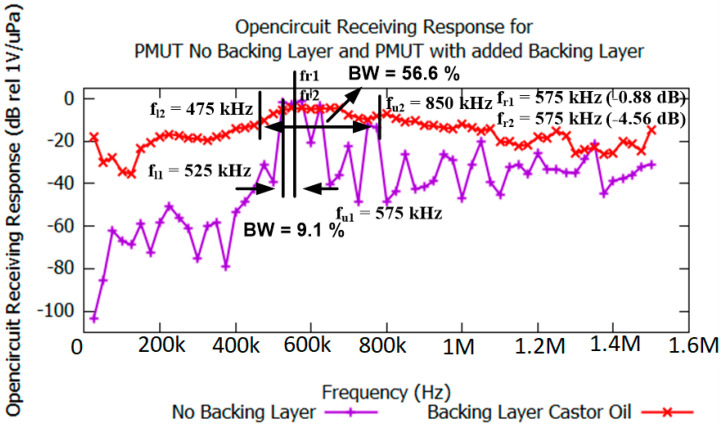
Open-circuit receiving responses for a PMUT with No Backing layer (single flexural membrane) and a PMUT with added Backing Layer Castor Oil (double flexural membrane structure with the same membrane material and electrode dimensions as the single flexural membrane).

**Figure 9 sensors-21-05582-f009:**
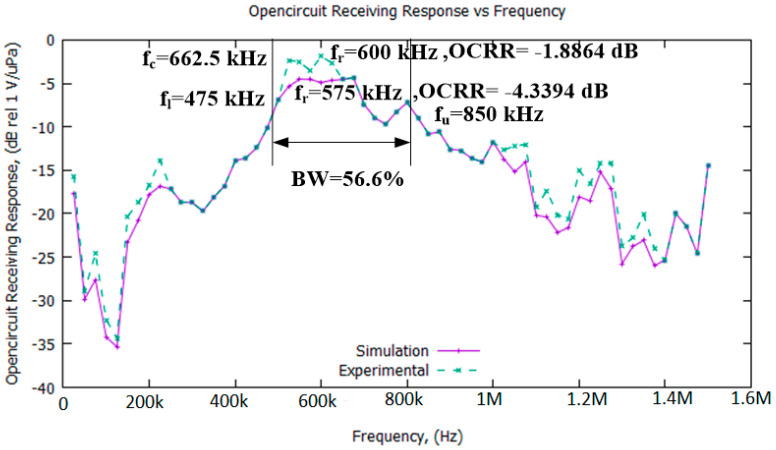
Open-circuit receiving responses for the proposed double-membrane device (simulated and experimental results).

**Table 1 sensors-21-05582-t001:** Parameters for all simulated PMUTs.

Material	Thickness (mm)	Width (mm)	Length (mm)
PDMS (Matching Layer)	0.1	6.5	14
Membrane	0.2	0.5	1.5
Electrode, Cu	0.0341	0.3	0.5
Polyimide	0.0458	6.5	14

**Table 2 sensors-21-05582-t002:** Parameters of the piezoelectric materials.

Piezoelectric Material	Mass Density, *ρ* (kg/m^3^)	Young’s Modulus, *E* (GPa)	Poisson’s Ratio, ν	Piezoelectric Coefficient, *d*_33_ (pm/V)	Dielectric Constant, K	Acoustic Impedance, *Z* (kg/m^2^ s)
Aluminium Nitrate, AlN	3250–3300	300–395	0.22–0.29	3.4–6.4	8.5–10.5	(33–36.3) × 10^6^
Lead Zirconate Titanate, PZT5H	7600	56–98	0.27–0.3	60–223	300–1300	34.2 × 10^6^
Polyvinylidene Fluoride, PVDF	1780	2.5	0.33–0.4	35	6–8	4.6 × 10^6^
Zinc Oxide, ZnO	5610–5720	208	0.36	5.9–12.4	10.9	(35.5–36.2) × 10^6^
Quartz, Q	2648	71.7	0.17–0.2	2.3 (*d*_11_)	4.3	13.3 × 10^6^

**Table 3 sensors-21-05582-t003:** Parameters of the final PMUT with double cavities and fluidic backing layers.

No.	Material	Functional	Thickness (mm)	Length (mm)	Width (mm)
1.	Polydimethylsiloxane (PDMS) top site	Top layer/acoustic matching layer	0.100	7.500	1.500
2.	Lead zirconate titanate (PZT5H)	1st membrane	0.200	1.500	0.500
3.	Positive/ground electrode	Electrode	0.034	0.458	0.215
4.	Polyimide	Substrate for 1st structure and membrane for 2nd layer	0.046	13.504	3.500
5.	Castor Oil	Backing layer	1.000	3.005	2.005
6.	Polydimethylsiloxane (PDMS)	Fluidic container	4.000	15.002	4.005

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
