# Peer review of "A Fluidics-Based Double Flexural Membrane Piezoelectric Micromachined Ultrasonic Transducer (PMUT) for Wide-Bandwidth Underwater Acoustic Applications"

_sensors, 2021, doi:10.3390/s21165582_

Round 1
Reviewer 1 Report
This manuscript looks like a technical report or user manual. The authors need to convince the reviewers the advantages of the proposed devices over state of the art devices( techniqes).
Author Response
Dear Reviewer,
Thanks for fruitful comment and really appreciate for review our paper, in improve the article. The feedback as attachment.
Thanks

Reviewer 2 Report
Authors fabricated new PMUT devices for wide bandwidth underwater applications. The measured sensitivity of -1.88 dB and 28.3 % fractional bandwidth are good index. Authors also showed detail fabrication process and simulated FEA analysis.
The article showed some novel structures so it is worthwhile to be published. However, authors need to correct some wrong words and add some missing references before publishing the article. Therefore, the article could be minor revision if authors follow the guidelines.
1. In Line 26, re -> and.
2. In Line 27, 600kHz and 662.5kHz -> 600 kHz and 662.5 kHz.
3. Correct , red to , black font.
4. In Figure 2, please check PZT5H color. Inside the schematics, it looks like PDMS.
5. In Figure 3, fonts need to be large.
6. Figure 5b quality needs to be improved.
7. In Line 262, 1325kHz -> 1325 kHz.
8. In Line 263, 575kHz -> 575 kHz.
9. Please provide some missing information about the city, country, and date for conference papers.
10. Please provide the reference ( which generates the 40 electric charge to induce an electrical signal in response to the amount of acoustic energy hitting the membrane) with the reference (Kim, K., & Choi, H. (2021). High-efficiency high-voltage class F amplifier for high-frequency wireless ultrasound systems. PloS one, 16(3), e0249034. )
11. Please provide the reference (The fractional bandwidth can be calculated by the equation ) with the reference (Shung, K. Kirk, and Gary A. Thieme. Ultrasonic scattering in biological tissues. CRC press, 1992. ) or another reference.
12. In FIgure 5b, input connection from function generator seems to be wrong. Please check the connection.
13. In Figure 2, authors selected air cavity with 0.0278 mm. Is there any reason ? It seems to be large.
14. In Figure 3, from the matching layer to the fluidic cavity looks like vertical structure. It is quite hard to have exact height. Is there any side effects with acoustic wave ?
15. In Figure 4, injected castor oil in fluidic channels need to be filled without any empty space. Is there any problems if castor oil is not fully filled ?
Author Response
Dear reviewer,
Thanks for your comment and really appreciate your comment in improvement our article. The feedback as attachment.
Thanks

Reviewer 3 Report
The authors propose a new structure of a double flexural membrane with a fluidic backing layer based on an in-plane polarization mode to optimize both the receiving sensitivity and frequency bandwidth, for underwater acoustic use. They the membrane material and the electrode configuration They simulated some piezoelectric membrane materials and various electrode dimensions by means of finite element analysis The optimized designed was manufactured demonstrating to have an improved bandwidth of 28.3% with the receiving sensitivity as high as -1.8864 dB re 1V µPa. The central frequency (around 600 kHz) was suitable for the targeted range of frequency. The paper is a good work in which, both simulated and experimental methodologies have been combined to achieve the demonstrated improvements.
-Could you include relevant distances in Figure 1?
-Please to define PDMS and PZT5H in the caption of Figure 1
-Equation 1. No conventional symbols are used. Why? For example, impedance typically is Z.
-Equation 2. Please to write in a more formal way from the mathematical point of view. Remove 1 and says that the units of OCRR have to be V/uPa. Alternatively, include OCCRo when OCCRo=1 V/uPa.
-Figure 2. Please, include all relevant distances.
-Table 2. There is mistake. Please, remove the remaining “)”
-Figure 3. Please to include all the distances. Picture quality is very low.
-Figures 6 and 7. The letters in the both axis are really small, and cannot be watched
-Conclusion section: You should include the main quantitative conclusions.
Author Response
Dear Reviewer,
Thanks for fruitful comment, and really appreciate in improve our article. The feedback as attachment.
Thanks

Round 2
Reviewer 1 Report
I did not see any improvement according to my previous comments.
Reviewer 3 Report
Paper is now available for publication since most of the inquires have been taken into consideration.